# Channel Permutations for N:M Sparsity

**Jeff Pool**
NVIDIA
jpool@nvidia.com

**Chong Yu**[1,2]
[1]Academy for Engineering and Technology, Fudan University
[2]NVIDIA
chongy@nvidia.com

## Abstract

We introduce channel permutations as a method to maximize the accuracy of N:M sparse networks. N:M sparsity requires N out of M consecutive elements to be zero and has been shown to maintain accuracy for many models and tasks with a simple prune and fine-tune workflow. By permuting weight matrices along their channel dimension and adjusting the surrounding layers appropriately, we demonstrate accuracy recovery for even small, parameter-efficient networks, without affecting inference run-time. We also present both a quality metric to simplify judging permutations as well as efficient methods to search for high-quality permutations, including two optimizations to escape local minima. Finally, we share an ablation study to show the importance of each part of our search algorithm, experimental results showing correlation between our quality metric and final network accuracy, improved sparse network accuracy using our techniques with insignificant overhead to training time, and the transformation of unstructured to structured sparse workloads. Code to use these techniques when generating a 2:4 sparse network is available at https://github.com/NVIDIA/apex/tree/master/apex/contrib/sparsity.

## 1 Introduction

Deep Neural Networks (DNNs) excel at complex tasks such as image classification, object detection, and language modeling. Their success often comes at the cost of immense computational complexity, and growing model size generally results in even higher-quality results. To combat this ever-increasing computational workload at inference, various approaches have been used to save effort; for example, network pruning removes weights from the network, and quantization can replace floating-point operations with simpler integer operations. We focus on one type of network pruning that is accelerated in hardware: 2:4 structured sparsity [22] (a particular form of the more general N:M sparsity), since it has been shown to maintain accuracy across a wide range of networks and tasks while being easy to save computation. While many networks do recover accuracy with a fine-tuning step, some small, parameter-efficient image-classification networks that were designed with economy of memory and computation in mind cannot recover the accuracy lost due to pruning with this suggested workflow. Further, this workflow repeats the dense training all over again, which, while straightforward, can be costly if the lifetime of the deployment model is not long enough to amortize this extra training.

We seek to address these shortcomings with the observation that the 2:4 pruning step, which forces two out of every four consecutive elements to be zero (the "2:4 constraint"), sometimes must prune a relatively large, and intuitively important, value. If different options were available to each group of four values, then more advantageous choices can be made, as illustrated in Figure 1. When the example weight matrix (top-left) is pruned with 2:4 sparsity in its original order, it results in a sparse matrix (top-right) with total weight magnitude of 44.5. By first changing the order of the columns in the matrix, though (bottom-left), the resulting sparse matrix (bottom-right) has a total weight magnitude of 58.6. It is simple to find this fruitful column permutation in such a small example, but

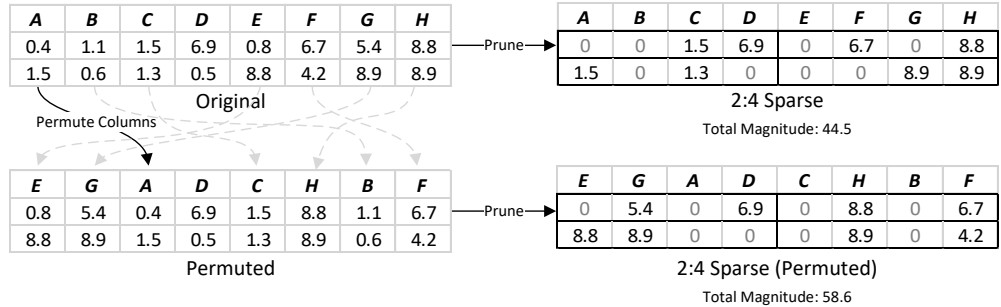

Figure 1: Permuting weights before pruning them helps preserve weight magnitude.

the complexity of the problem explodes with problem size. Further, once a permutation is found, applying it some layer of a DNN can result in a performance overhead at run-time [14]. In this work, we give solutions to both these problems and show several benefits of using permutations for N:M sparsity. While we focus on a particular variant of N:M structured sparsity for practical reasons (acceleration with readily-available hardware [26]), the techniques and observations herein apply equally to all N and M. Similarly, our permutation search strategies can be used to increase or decrease any metric of interest, not just magnitude.

We note that while weight magnitude is not proven to be the optimal pruning metric, and that increasing magnitude is not guaranteed to improve accuracy, we show it to work well for the cases studied herein.

Our contributions include (1) a method of permuting input channels of weight matrices without changing the output of the network or incurring a performance overhead, (2) an efficient algorithm for generating unique permutations for N:M structured sparse matrices, (3) two methods to overcome a shortcoming of greedy permutation generation algorithms, (4) searching for and applying permutations to improve accuracy of 2:4 sparse networks, and (5) transparently transforming unstructured sparse to structured sparse matrices.

The rest of this paper is as follows: Section 2 discusses previous work in the area and provides background information on matrix permutations, and Section 3 presents our method of to permuting weights of a neural network with no run-time overhead. In Section 4, we present a quality metric, an intractable exhaustive solution to find a high-quality permutation, a greedy solution, and improvements to the baseline greedy algorithm. We share an ablation study, quality metric/accuracy correlation, accuracy improvements, and results of transforming unstructured to structured sparsity in Section 5, along with a brief discussion of the runtime needed to search for permutations. Finally, we offer a discussion and areas for future work in Section 6.

## 2   Related Work and Background

**Network pruning.** Removing weights from a network is used to reduce both the storage and bandwidth required for the network as well as the computational power required to deploy the network. The area of network pruning is multi-dimensional: one can prune to different amounts, with different sorts of structure or regularity, using weight magnitude or other metrics to choose which weights to make zero [23, 16], pruning gradually [6, 33] or in one shot, keeping the network the same size or growing it to accommodate for the sparsity [5], etc.

Let us focus on structure and regularity; this dimension is the most important for this work and is generally composable with the others. Early work in pruning DNNs used unstructured sparsity, removing individual weight values [6], but research soon moved on to pruning at a coarser granularity [34, 1, 19, 18, 15]: at the cost of accuracy loss for the same amount of sparsity [21, 38], removing entire channels or filters results in a smaller, dense workload which is easy to accelerate. At the extreme, sparsity can be in the form of blocks [5] or entire layers [4]. Simply put, network pruning has been a balancing act between maintaining accuracy and inducing enough sparsity to impart a useful benefit, either in storage requirements or the runtime of the resulting model.

**N:M structured sparsity.** Recently-released hardware accelerates matrix-multiply-accumulate instructions if one operand satisfies a 2:4 sparsity constraint [22], a particular form of N:M sparsity in which at least N out of every M (contiguous, aligned) elements are zero. In particular, this hardware requires the sparsity to be applied to the input channel dimension, $C$, of convolutional layers: weights along this dimension are divided into groups of four weights, and two of them are forced to zero. A workflow to generate networks that satisfy this 2:4 constraint, using a form of Learning Rate Rewinding [28], has been empirically verified to maintain accuracy across a wide range of networks and tasks (with the exception of the aforementioned small, parameter-efficient networks).

Hubara et al. report a technique for finding good transposable masks for use in both forwards and backwards passes [11], as well as two techniques for improving accuracy when fine-tuning N:M sparse networks for inference, but they are orthogonal to this work. Though we focus on fine-tuning an existing dense network, there has been work in training N:M networks from scratch [37], but results for 2:4 sparsity fall short of the dense baseline.

**Matrix permutations.** Applying a permutation to a matrix changes the order of the individual rows or columns that make up the matrix, and can be specified as a vector $P$ of indices, with no repetitions, from 0 to the size of the dimension (-1). These indices are used to permute the matrix by moving the column (or row) at index $j$ to position $P[j]$.

A matrix multiplication's operands can be permuted in two ways without materially affecting the results: permuting the inner dimension or the outer dimension of the operands. Simultaneously permuting the inner dimension of both operands changes the *order* in which the partial sums of the dot-product are computed, but not the eventual sum in each position of the output matrix. An example of this appears in the right side of Figure 2: Layer $i$'s input channel, or $C$, dimension, common to both operands, is permuted (numbers 1 and 2) by permutation $P_i$, and the output is unaffected. On the other hand, permuting rows of the first operand (number 3 on the left side of Figure 2) or the columns of the second operand, either outer dimension, affects the position of the corresponding rows or columns in the output matrix. For this reason, when rows are permuted in the weight matrix of a neural network, it is necessary to apply the inverse permutation to the output to restore the original order. Since we are concerned with the order of only the $C$ dimension, which defines each 2:4 group, permutations will be applied to the *inner* dimension of GEMM operations in high-performance math libraries [3]. Herein, permutations will apply to columns of a matrix's weights or channels of a convolution tensor's weights interchangeably. Each layer (or, as discussed in Section 3, group of layers) can have a unique permutation, $P_i$ (for layer $i$).

These properties have been used to cluster large values together when pruning a network with group sparsity, such as blocks or vectors [14]. The approach in this prior work has two shortcomings: it (1) incurs a runtime performance penalty and (2) can not escape from poor local minima. Permutations have also been used to make unstructured sparsity easier to accelerate on systolic arrays [2].

## 3 Applying Permutations

### 3.1 Permutations Across Layers

To apply a desirable permutation to some layer $i$'s $C$ dimension, we can permute its weights just once, offline. To avoid an explicit operation to apply the same permutation to its input activations, we exploit the property that permuting weights along the output channel, or $K$, dimension also permutes output activations along the $K$ dimension, and the *output* channel dimension of one layer's activations is the *input* channel dimension of those same activations when used as the input to the next layer.

Consider permutation $P_i$ desired by layer $i$: that layer's weights will be permuted along $C_i$, as indicated by the '1' on the right side of Figure 2. In order to keep the output constant, its activations need the same permutation applied along $C_i$ ('2'). We effect this input activation permutation with another offline step: permuting the previous layer's weights along $K_{i-1}$ ('3'). By permuting the $K$ dimension of layer $i - 1$'s weights to reorder its output activation's $K$ dimension, we have prepared layer $i$'s input activations as required by $P_i$ without an explicit permutation operation. Since each layer needs to permute its own weights along only one dimension and can "absorb" an orthogonal permutation in the other dimension without conflict, we incur no performance overhead at run-time. We note that this simplification could also be applied to past work to eliminate the

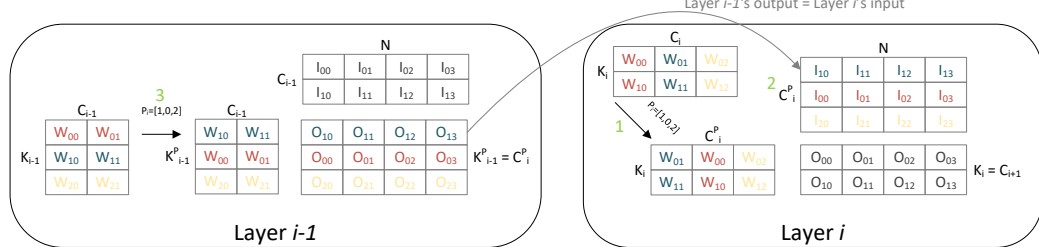

Figure 2: Permuting along the inner dimension of a GEMM (Layer $i$, right side) does not change the output, but permuting along an outer dimension (such as Layer $i-1$'s rows, left side) causes a corresponding permutation in the same dimension of the output. We exploit these two properties to apply some desired permutation $P_i$ to Layer $i$'s weights ('1'). The required matching permutation in the same dimension of the input activations ('2') is effected by permuting the previous layer's weights along rows using the same permutation ('3'), requiring no run-time overhead. Colors show the order of rows (activations) and columns (weights) involved in each GEMM operation.

reported performance overhead due to each layer's desired permutation being applied in multiple dimensions [14], potentially at the expense of final network accuracy.

So, each layer may see two permutations: one that is for itself, applied along its $C$ dimension, and one that is desired by a downstream layer (or layers, below), applied along its $K$ dimension. For example: if following layer $i+1$ has its own desired $C$ permutation, layer $i$ would be responsible for permuting the activations by permuting its own weights' $K$ dimension. This second permutation is orthogonal to and will not conflict with its own $C$ dimension permutation. Similarly, layer $i-1$ can apply a permutation to its $C$ dimension without affecting the $K$ permutation for layer $i$.

### 3.2 Handling Complexities of Deep Neural Networks

**Forks and joins.** Many network types, including Residual Networks [7], have forks and joins in their network graphs, leading to layer being a "parent" to multiple layers in that its outputs are inputs to some number of downstream layers (children). Since a parent layer can only have one output channel order, all children of a parent must share the same input channel order: the children's permutations are constrained to be identical. Forcing the same permutation to be used for multiple layers is straightforward: concatenating the weight matrices along the input channel dimension (which, if coming from a fork, will be of the same size), searching for a good permutation (see Section 4), and splitting the earlier concatenation will force multiple weight matrices to have the same permutation that has been optimized for all layers. A layer with multiple parents (after a join, for example), can apply its input channel permutation, once found, to its parents' output channel dimension.

**Pass-through operations.** When applying a permutation to a parent, we can stop only when there is a GEMM operation (convolution, FC layer, recurrent layer, etc.), which can apply the permutation to its weights' outer dimension to prepare the output activations. Until we encounter that operation, we must apply the permutation to the output channel dimension of the current operation then pass the permutation through to the next parent. For example, a convolution's outputs are generally subject to a bias term, BatchNorm [13], and an activation function before they are input to the next convolution operation. Bias and BatchNorm operations have per-output channel parameters, thus they require the permutation to be applied to keep the network graph consistent with the originally parameterized function. Activation functions are typically performed per-element and have no notion of a channel dimension, so these types of layers do not need to apply the permutation before passing it through to their parent. Quantizing a neural network typically involves applying scaling factors to weight tensors, often per-channel [35]; these factors are permuted before the permutation is passed on.

**Input layers.** For most layers in a network, the activation permutation can be accomplished by permuting the preceding weights, as above. The first layer cannot use this technique, though, since there are no preceding weights. We choose to forego permutations on the input layers, which is generally not a noticeable decision since (1) input layers are often not pruned for accuracy reasons, and (2) networks use for image-based tasks, like classification and object detection, have only three input channels, which are too few to be accelerated with 2:4 sparsity [22].

# 4 Finding Good Permutations

## 4.1 Efficacy: What Makes a Good Permutation

To avoid fine-tuning the network for every potential permutation, we construct a quality metric based only on the input and permuted matrices. We use weight magnitude to choose weights to prune, so it is natural to maximize the total magnitude after the 2:4 constraint's application: a perfect permutation would preserve all the magnitude of the input matrix. In order for a perfect permutation to be achievable, the input matrix must have at least 50% sparsity in each row. The magnitude after enforcing 50% sparsity in each row, $|W_{50\%\_rows}|$, is an upper bound for a permutation's quality. We define a metric, **efficacy**, to quantify the quality of a permutation $p$ applied to a weight matrix $W$ ($W^p$ when permuted), in terms of how much it improves the magnitude with respect to this upper bound:

$$efficacy = 1.0 - \frac{|W_{50\%\_rows}| - |W^p_{2:4}|}{|W_{50\%\_rows}| - |W_{2:4}|}$$

If the magnitude after the 2:4 constraint of the original and permuted matrices are the same, then the permutation changed nothing and has an efficacy of 0.0. On the other hand, if the magnitude after the permutation is the same as that of the 50% row sparse matrix, then the efficacy metric will be a perfect 1.0 and it will be clear that no better permutation can be found. Finally, if the magnitude of the permuted matrix is worse than the original matrix, the permutation's efficacy will be negative.

## 4.2 Exhaustive Searches

**Intractability.** For a network layer with $C$ channels, there are $C!$ channel permutations possible; this is nearly 21 trillion permutations for only 16 channels. However, there are two characteristics of N:M structured sparse matrices that reduce the number of *unique* permutations to only $\frac{C!}{M!^G G!}$, where $G = C/M$, the number of pruning groups, or "stripes" of the matrix. A stripe is a vertical slice of a matrix covering exactly M (aligned) columns, and the first simplifying observation is that the order of the columns within a stripe is immaterial; the same pruning decisions will be made regardless of this order, as long as the stripe contains the same columns. The second observation is that the order of stripes in the larger matrix does not matter, since pruning decisions are local to a stripe. For a layer with 16 channels and targeting 2:4 structured sparsity, the number of unique permutations is reduced to only 2.6 million. However, this value still grows much too quickly to test them all for even moderate channel counts: at 32 channels, there are nearly 60 quintillion unique permutations. (See Appendix C for details.)

**Algorithm.** There are too many unique permutations to check beyond a small number of channels, but smaller matrices can still be exhaustively searched (we will exploit this in Section 4.4). Here, we design an algorithm to generate only the unique permutations from the set of $C!$ total permutations.

First, we define a canonical form for a unique permutation, exploiting the two observations from the previous subsection: a permutation is unique only if each of its stripes' channels are in sorted order *and* the stripes are sorted with respect to each other (e.g. by the first value of each stripe). If two permutations result in the same pruning decisions, they will have the same canonical form. From these two restrictions, it can be seen that (1) all unique permutations start with column 0, and (2) if a column $c$ begins a group, then every column $< c$ must be a member of a previous group.

Next, we use a standard recursive solution to the general problem of generating all permutations. We provide the permutation $[0]$ and remaining columns $[1..C-1]$ as input, and in turn move each entry from the remaining columns to the end of the current permutation. Each time the current permutation grows, the function is called again with the larger permutation and smaller list of remaining columns as input. Eventually, there will be no entries left in the remaining columns, and a full permutation can be added to a global list of permutations.

To limit this algorithm to only unique permutations, the recursive step is performed only if the current permutation is in canonical form and, therefore, unique. By checking at every stage of the generation algorithm, we can eliminate unnecessary effort and complete in linear time with respect to the number of unique permutations. (Listing 2 in Appendix C shows this algorithm.)

**Algorithm 1:** "Deep" greedy permutation search with bounded regressions to escape local minima

```
def Find_Permutation(matrix, num_cols, stripes_per_group=2, escape_attempts=100):
    permutation = [c for c in range(0, num_cols)];                    #Identity permutation
    for escape_attempt in range(escape_attempts+1, 0, -1):
        #Greedy phase:  optimize stripe groups that give large benefits
        while True:
            optimizations = Optimize_Stripe_Groups(matrix, permutation, stripes_per_group);
            optimization = Find_Largest_Positive_Improvement(optimizations);
            if optimization is None: break;
            permutation = Permute(permutation, optimization);
        #Escape phase:  attempt to escape the local minimum
        if escape_attempt > 1:
            src, dst = Unique_Random_Integers(num_cols);
            permutation = Swap_Columns(permutation, src, dst);
    return permutation;                                              #Final permutation
```

## 4.3 Greedy Incremental Improvements

We begin building an efficient permutation generation algorithm by adapting a greedy algorithm used for group-sparse patterns [14]. At each iteration of the base algorithm, a sparsity mask is first determined to maximize weight magnitude. Then, all pairs of channels are speculatively swapped; only the swap that leads to the greatest increase in magnitude for the mask found in step one is enacted, generating a new permutation and concluding a single iteration. The algorithm ends when there is no swap that increases magnitude. This approach has useful qualities: it provably converges using greedy decisions at each step, it can be parallelized, and it can be optimized to minimize the cost of the $C^2$ potential channel swaps considered at each step. However, (1) it does not directly fit the problem of 2:4 sparsity, and (2) it can get trapped in a local minimum.

Since there is no notion of blocks/vectors of zeros and nonzeros and our goal is not simply to cluster together values with large magnitudes, splitting the mask-selection and permutation-selection into separate steps is fruitless. Instead, it is better to consider the quality of a potential permutation as though the new mask were determined *after* the permutation, rather than trying to make a permutation conform to a given mask. For N:M sparsity, this can be simplified: since each stripe is independent from every other stripe, the benefit to the entire matrix of swapping two channels is the same as the benefit to the two stripes to which those channels belong.

This modified greedy algorithm converges rapidly, but it is highly likely that this convergence will be to a local minimum. We can show this empirically with matrices that are small enough to be solved exhaustively: Table 1 shows that the basic channel-swapping greedy phase finds the optimal solution for twenty-five 32x16 matrices (which are small enough to be solved exhaustively) only three times. As the input matrix grows, it is increasingly likely that the solution will be a local minimum.

## 4.4 Escaping Local Minima

Here, we introduce two techniques we use to escape local minima *without* the possibility of falling into a worse solution. (Appendix E discusses some ways to change, but not necessarily improve, the minimum found; such techniques can be applied on top of those presented in this section, which provably do *not* worsen an already-converged solution.)

**Escape technique 1: bounded regressions.** After convergence, swapping any two channels $a$ and $b$ will cause a regression in the permutation's efficacy. However, after repeating the greedy process until convergence, one of two things may happen:

1. There is no pair of channels that result in an improvement greater than the *reduction* caused by the swap of $a$ and $b$. In this case, $a$ and $b$ are swapped back to their original position by the greedy step and the "new" solution is the original converged solution.
2. There *is* some pair of channels $c$ and $d$ that result in an improvement greater than the reduction of the swap of $a$ and $b$. The solution after $c$ and $d$ are swapped by the greedy process will be better than the original solution.

Since perturbing a solution by swapping any two channels and iterating to convergence will never reduce the efficacy of an already-converged solution, we call these perturbations "bounded regressions." To prove the current solution's minimum is at least two swaps away from a better solution, we can swap every pair of channels exhaustively upon convergence, but in practice we swap two random channels, up to $B$ times. *Any* two columns can be swapped, but the search horizon is limited to a depth of only one channel swap, so these bounded regressions are broad, but shallow. We must add depth to the search horizon to find solutions that are two or more swaps away.

**Escape technique 2: narrow, deep search.** We add depth to our search process by exhaustively optimizing stripe groups: choosing $D$ stripes to form a group and then exhaustively searching for the best permutation for that group in isolation. Optimizing a group of two stripes ($D = 2$) is trivial since there are only 35 unique permutations, but this allows an effective search depth of 2: the best permutation in for a group of 8 channels may be two swaps away from the initial order. In some cases, $D = 3$ (or $D = 4$) is still feasible, with 5,335 (2.6 million) unique permutations, for a search depth of 4 (8). We use our O(N) algorithm from Section 2 to generate only the unique permutations from the set of over 479 million (nearly 21 trillion) potential permutations. Similar to bounded regressions, optimizing stripe groups as a method to escape a local minimum can be as simple as randomly choosing $D$ stripes, optimizing, and then repeating the greedy phase. Since these stripe groups are optimized atomically, there is no chance for the overall solution to worsen.

**Putting it all together** Our final column permutation generation algorithm, shown in Algorithm 1, maintains the iterative greedy step as phase one. However, each greedy step consists not of swapping two columns for maximum benefit, but exhaustively optimizing some number of stripes (we use $D = 2$ unless otherwise specified). Every potential group of $D$ stripes, $\binom{\#stripes}{D}$ in total, is speculatively optimized, then the group with the most beneficial permutation is chosen, then all stripe groups are speculatively optimized again, and so on. This modification incorporates the narrow, deep searches into the greedy phase, helping to bypass some local minima without first converging to them.

We use bounded regressions to try to escape the local minimum found by phase one. Phase two is instantaneous: two random channels are swapped. The real escape is up to the greedy phase, which attempts to improve upon the permutation that resulted from the swap. If the greedy phase escapes the previous minimum, it will repeat to converge to some new solution, else it will return to the previous minimum, and the escape phase will repeat again; we default to 100 total escape attempts.

## 5   Experiments

**Ablation study.** We examine the benefit of each technique used to escape local minima. As a baseline, we generate 1000 random permutations and choose the best. We then show two types of greedy phases: the baseline of simply swapping channels, and the enhanced option of optimizing stripe groups. When optimizing stripe groups, they can be comprised of either two or three stripes (8 or 12 total channels); the higher number affords a higher search depth at the cost of higher search time. We also examine using bounded regressions as a local minimum escape phase, allowing up to 100 total attempts to escape the various minima found during the greedy phase. We report the average efficacy for a set of 25 random 128-column, 64-row matrices in Table 1. Both enhancements improve on the baseline greedy approach; while the bounded regressions increase efficacy whenever they are used, increasing the number of stripes per optimization group provides a larger benefit when used in the greedy phase. We also show that for a set of 25 random 32x16 matrices, which can be solved exhaustively to find the optimal permutation, $p_o$, our techniques find $p_o$ much more often than the baseline, and both techniques together improve even further.

**Efficacy/accuracy correlation.** Having shown that our techniques improve the efficacy metric over a baseline approach adapted to this task, we now show the utility of this metric by demonstrating a strong correlation between permutation efficacy and final accuracy of the fine-tuned network. We use the ILSVRC12 [29] dataset to evaluate SqueezeNet v1.0 [12] and EfficientNet B0 [32], which are trained densely before being pruned and fine-tuned according to the procedure [22] implemented in the ASP [25] library using PyTorch [27] and NVIDIA V100 [24] accelerators. By first finding the best permutation using the technique in Section 4.4, we can then generate permutations that evenly divide the efficacy into eighths, from 0 up to the best permutation's efficacy for each layer. Fine-tuning each of these new seven intermediate networks produces the curve of accuracy improvement vs. efficacy shown in Figure 3. Both networks' final accuracy increases monotonically with permutation

Table 1: Ablation study: escaping local minima with bounded regressions and using a more thorough greedy phase. BR ($B$) = bounded regressions (a total of $B$ pairs of channels are swapped when the greedy phase converges), OSG ($D$) = optimize stripe groups of size $D$ during the greedy phase. Search times were collected on a V100 accelerator.

| | | $25\times$ 64x128 | $25\times$ 32x16 |
|---|---|---|---|
| **Greedy Phase** | **Escape Phase** | **Efficacy (%)** | **Optimal Found (#)** |
| Random (1000) | - | 8.0 $\pm$3.0 | 0 |
| Channel Swap | - | 46.5 $\pm$2.2 | 3 |
| Channel Swap | BR (100) | 47.1 $\pm$1.9 | 11 |
| Channel Swap | BR (1000) | 49.1 $\pm$1.9 | 11 |
| OSG (2) | - | 47.7 $\pm$2.2 | 7 |
| OSG (2) | BR (100) | 48.2 $\pm$2.2 | 15 |
| OSG (2) | BR (1000) | 49.5 $\pm$1.9 | 16 |
| OSG (3) | - | 51.9 $\pm$1.7 | 17 |
| OSG (3) | BR (100) | 52.3 $\pm$1.6 | 23 |
| OSG (3) | BR (1000) | 53.4 $\pm$1.8 | 25 |

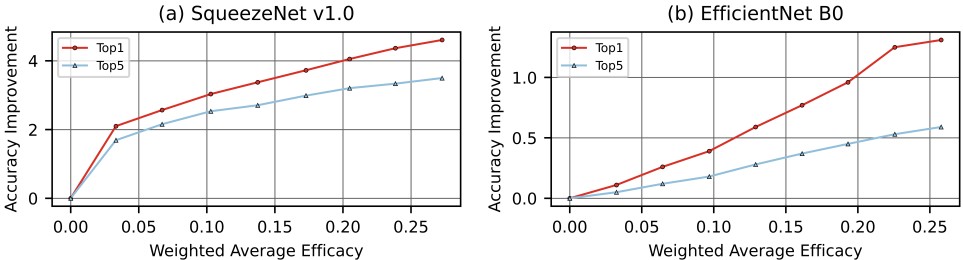

Figure 3: Network accuracy increases monotonically with permutation efficacy.

efficacies (the weighted average over all weight layers is used as the x-axis). SqueezeNet v1.0 sees a large improvement after even (relatively) poor permutations are used, while EfficientNet B0 reflected a steady improvement through the entire range of efficacies.

**Network accuracy.** The primary goal of our technique is to allow the suggested workflow [22] to maintain accuracy for parameter-efficient networks. We demonstrate the success of channel permutations on several such networks. First, we train each network densely from scratch, using hyperparameters from public repositories (see Appendix A). Then, we prune the dense models with 2:4 sparsity and repeat the training process again, resetting hyperparameters and optimizer state [22]. Finally, we search for and apply permutations (using the technique from Section 4.4) to each layer of the dense models, *then* prune and repeat the training process to fine-tune the remaining weights.

Accuracy results are shown in Table 2: the absolute top-1 and top-5 accuracy values are shown for the dense baseline network, then the difference in the default-pruned and permuted-pruned models with respect to the dense baselines are shown in the third pair of result columns. In every case, the accuracy of the network that was permuted before pruning is higher than pruning the network in its default channel order. Moreover, for the small models that cause trouble for the default process, our technique yields models with accuracy that is equivalent to the dense baseline model. Finally, to show the universality of our technique, we include results for larger models that did not struggle with the default pruning order; even for these networks, accuracy is improved. This suggests that permutations may be safely applied to all networks, not just those that fail without permutations, and also that the number of fine-tuning samples necessary to recover accuracy may be reduced by using permutations. Appendix B shows the same trends for other tasks: semantic segmentation, object detection, language translation, and language modeling.

We perform one extra accuracy study, fine-tuning MobileNet v2 and MNASNet 1.0 after applying permutations found by the baseline Channel Swap strategy. Their final top-1/top-5 accuracies remained worse than the dense baselines by -1.26/-0.69 (MobileNet v2) and -0.66/-0.51 (MNASNet 1.0). While this is an improvement over pruning in the default order, predicted by weighted average

Table 2: Using the same fine-tuning process, channel permutations improve accuracy for all networks and incur only a trivial increase in training time. Training and fine-tuning uses 8 V100 accelerators, the permutation search uses only one V100. (ILSVRC12)

| Network | Baseline (Dense) | | Default 2:4 | | Permuted (ours) | | Search Time | |
|---|---|---|---|---|---|---|---|---|
| | Top1 | Top5 | $\Delta$Top1 | $\Delta$Top5 | $\Delta$Top1 | $\Delta$Top5 | mm:ss | % Train+FT |
| MobileNet v2 [30] | 71.55 | 90.28 | -1.98 | -1.13 | 0.01 | 0.02 | 00:50 | 0.02% |
| MobileNet v3 (Small) [9] | 67.67 | 87.40 | -2.73 | -1.67 | 0.10 | 0.15 | 00:32 | 0.01% |
| MobileNet v3 (Large) [9] | 74.04 | 91.34 | -0.91 | -0.40 | 0.10 | 0.04 | 00:58 | 0.01% |
| SqueezeNet v1.0 [12] | 58.09 | 80.42 | -4.01 | -2.96 | 0.60 | 0.54 | 00:12 | 0.02% |
| SqueezeNet v1.1 [12] | 58.21 | 80.62 | -1.25 | -0.93 | 0.03 | 0.05 | 00:12 | 0.02% |
| MNASNet 1.0 [31] | 73.24 | 91.36 | -1.25 | -0.58 | 0.02 | 0.00 | 01:10 | 0.04% |
| ShuffleNet v2 [20] | 68.32 | 88.36 | -1.35 | -0.87 | 0.10 | 0.01 | 00:14 | 0.01% |
| EfficientNet B0 [32] | 77.25 | 93.59 | -1.27 | -0.52 | 0.04 | 0.07 | 01:34 | 0.04% |
| ResNet-50 [8] | 76.16 | 92.88 | 0.05 | 0.13 | 0.13 | 0.26 | 02:13 | 0.10% |
| ResNeXt-50 [36] | 77.62 | 93.70 | 0.01 | 0.05 | 0.13 | 0.07 | 02:57 | 0.08% |
| DenseNet-161 [10] | 77.14 | 93.56 | 0.82 | 0.41 | 0.92 | 0.52 | 00:56 | 0.02% |

Table 3: Runtimes for various strategies searching for good permutations for various input sizes show that our improved search algorithm completes quickly for even large matrices (default settings **emphasized**). Results captured on a single V100 accelerator.

| | Cols | 32 | | 64 | | 128 | | 256 | | 2048 |
|---|---|---|---|---|---|---|---|---|---|---|
| Strategy | Rows | 32 | 64 | 64 | 128 | 128 | 256 | 256 | 512 | 2048 |
| OSG(2) | | 0.00 | 0.00 | 0.01 | 0.01 | 0.02 | 0.03 | 0.17 | 0.23 | 29.99 |
| **OSG(2),BR(100)** | | 0.07 | 0.09 | 0.11 | 0.13 | 0.21 | 0.27 | 0.57 | 0.75 | 59.85 |
| OSG(2),BR(1000) | | 0.71 | 0.83 | 0.99 | 1.26 | 1.87 | 2.49 | 4.65 | 6.07 | 332.93 |
| OSG(3) | | 0.12 | 0.20 | 0.37 | 0.84 | 8.76 | 22.60 | 129.83 | 348.19 | - |
| OSG(3),BR(100) | | 3.26 | 5.92 | 8.29 | 14.59 | 68.35 | 135.76 | 628.45 | 1191.39 | - |
| OSG(3),BR(1000) | | 32.47 | 58.26 | 76.37 | 139.35 | 580.23 | 1095.28 | 4666.89 | 9455.46 | - |

efficacies of 22.098% and 29.145%, it shows that it is important to escape these local minima in order to recover as much accuracy as possible. Weighted average efficacies for our suggested setting of OSG(2),BR(100) are 22.472% and 29.394%. (Appendix F has more details.)

**Search Time** While the apparent complexity of the improvements over the baseline algorithm may seem to make the technique too expensive to use in practice, an efficient implementation can reduce the time required to perform the search for a good permutation to an insignificant fraction of the time required to train the model, as shown in Table 2. The total training and fine-tuning time is never increased by more than one tenth of percent in our test networks. There are likely opportunities to improve the performance further, but the results on random matrices of various sizes in Table 3 show that the proposed settings (OSG(2),BR(100)) are fast enough for even large weight matrices.

**Unstructured to structured sparsity.** Unstructured pruning is commonly used to reduce model complexity. However, aggressive sparsity is required to out-perform dense matrix and vector instructions, so the achieved performance from unstructured sparsity is typically limited [11]. Using channel permutations, we can transform unstructured sparse layers into layers that conform to 2:4 structured sparsity so that they can take advantage of sparse matrix instructions [22]. From our accuracy results above, it is clear that we can reduce the impact of enforcing the 2:4 constraint on an already-sparse network to improve the fine-tuned accuracy, similar to prior work [11], but we can also *transparently* transform layers to structured sparsity if the permutation allows the weight values to remain as they are; if we make no extra zeros, then there is no fine-tuning necessary. To show the benefit of channel permutations for this purpose, we begin with a pre-trained ResNet-50 from torchvision.[1] By imposing different amounts of unstructured sparsity on this model's layers, we can generate plausible patterns of zeros; even if the resulting network is not very accurate due to the lack of fine-tuning, the zeros in the weights are in reasonable places. Then, we search for permutations for each layer at each sparsity. If a permutation is found that results in the same number of nonzero values before and after enforcing

---

[1]https://pytorch.org/vision/0.8/models.html

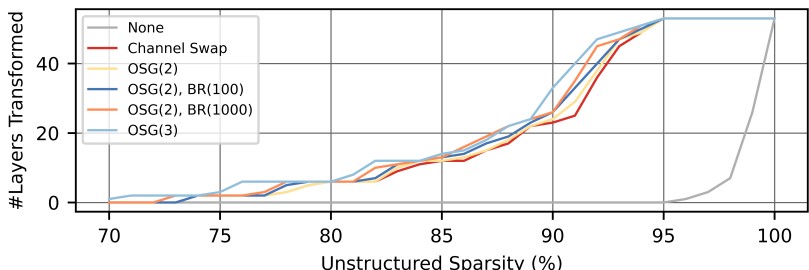

Figure 4: Permutations increase the number of unstructured sparse layers that can accommodate the 2:4 constraint without making extra zeros. Using deeper greedy searches (optimizing stripe groups instead of simply swapping channels) and escaping from local minima with bounded regressions further increase the number successfully-transformed layers.

the 2:4 constraint, then the layer can be transparently transformed. For each of six permutation search strategies, Figure 4 shows how many layers can be transparently transformed when each layer is pruned to different sparsities. Permutations allow more layers at each sparsity to be transparently transformed, and better permutations can further increase this transparently transformable layer count.

## 6    Conclusion

We propose the use of weight matrix permutations to reduce the impact of pruning with N:M structured sparsity, adapting prior work [14] to this new type of sparsity. Modifying the resulting greedy algorithm to use deeper searches by exhaustively optimizing groups of stripes and allowing bounded regressions help the search escape local minima. Our permutation quality metric, efficacy, shows a strong correlation with network accuracy improvements, and we show that small, parameter-efficient networks can use the existing fine-tuning workflow to recover accuracy after undergoing channel permutations. By limiting permutations to the inner dimension of a network's matrix operations, this technique requires only an offline step with no overhead when the model is deployed.

**Limitations.** While our results suggest that there may be a general workflow to recover accuracy that does not require as much overhead as the original dense training, we do not have evidence for such a workflow. Similarly, our heuristic of increasing weight magnitude empirically results in improved network accuracy, this may not hold in general; other heuristics may perform better.

**Potential societal impact.** Our technique aims to improve the accuracy of neural networks that require reduced resources to train and deploy. We do this by changing the order of weights along the channel dimension; a similar effect comes from using a different random seed to initialize the network, so societal impacts related to our technique are not readily apparent.

**Future work.** We see three areas for further study. First, permutations may reduce the number of fine-tuning samples required to recover the accuracy lost by pruning for networks that already succeed in maintaining accuracy. Second, there has been recent work in exploring how to train with N:M sparsity from scratch [37]. It may be possible to include permutations as part of this process by performing the greedy iterative process over time, amortizing the convergence of the permutation search over the convergence of the network. Also, to accelerate the data gradient calculation phase of the backwards pass, N:M sparsity in weights must be transposable: the N:M constraint must be satisfied on both rows *and* columns of the weight matrix. In this case, permuting rows may help reduce the impact of these transposable constraints at the cost of runtime overhead to permute activations between layers. Finally, we observed significant efficacy improvements when the depth of the searches was increased by optimizing more groups of stripes but quickly became limited by processing time. This time could be reduced by parallelizing the search over more processing resources or finding a more efficient implementation, perhaps by adapting Lehmer codes [17] to generate unique permutations on the processor where they will be tested. Formulating the problem for a combinatorial solver may also allow better solutions to be found.

## Funding Disclosure

This work was supported by Shanghai Municipal Science and Technology Major Project (No.2021SHZDZX0103); the Shanghai Engineering Research Center of AI & Robotics, Fudan University, China; and the Engineering Research Center of AI & Robotics, Ministry of Education, China.

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
