# Appendices

## A  Training Hyperparameters

Eight V100 GPUs were used for data-parallel training in every training experiment in the main paper. Other hyperparameters are detailed in Table 4.

Table 4: Training hyperparameters for the networks tested.

| Network | Optimizer | Initial LR | LR schedule | Momentum | Weight Decay | Epochs | Batch Size |
|---|---|---|---|---|---|---|---|
| (1) MobileNet v2 | SGD | 0.045 | Step (step_size=2) | 0.9 | 4e-5 | 300 | 32 |
| (1) MobileNet v3 (Small) | RMSprop | 0.064 | Step (step_size=2) | 0.9 | 1e-5 | 600 | 128 |
| (1) MobileNet v3 (Large) | RMSprop | 0.064 | Step (step_size=2) | 0.9 | 1e-5 | 600 | 128 |
| (1) SqueezeNet v1.0 | SGD | 0.04 | Linear | 0.9 | 2e-4 | 100 | 64 |
| (1) SqueezeNet v1.1 | SGD | 0.04 | Linear | 0.9 | 2e-4 | 100 | 64 |
| (2) MNASNet 1.0 | SGD | 1.0 | Cosine | 0.9 | 1e-5 | 300 | 256 |
| (1) ShuffleNet v2 | SGD | 0.1 | Cosine | 0.9 | 1e-4 | 400 | 128 |
| (3) EfficientNet B0 | RMSprop | 0.08 | Cosine | 0.9 | 1e-5 | 400 | 256 |
| (1) ResNet-50 | SGD | 0.1 | Step (step_size=30) | 0.9 | 1e-4 | 90 | 32 |
| (1) ResNetXt-50 | SGD | 0.1 | Step (step_size=30) | 0.9 | 1e-4 | 100 | 32 |
| (1) DenseNet-161 | SGD | 0.1 | Step (step_size=30) | 0.9 | 1e-4 | 90 | 32 |

Leading numerals indicate the repository used to train each model:

1. https://github.com/pytorch/vision/tree/master/references/classification

2. https://github.com/1e100/mnasnet_trainer/tree/master

3. https://github.com/NVIDIA/DeepLearningExamples/tree/master/PyTorch/Classification/ConvNets/efficientnet

## B  Results on Other Tasks

### B.1  Semantic Segmentation

We use TorchVision (https://github.com/pytorch/vision/tree/main/references/segmentation) and its default hyperparameters to perform experiments on several networks performing semantic segmentation on the COCO2017 dataset, shown in Table 5.

Table 5: Semantic segmentation networks improve under the same fine-tuning schedule when using permutations. (COCO2017)

| Network | Baseline (Dense) | | Default 2:4 | | Permuted (ours) | |
|---|---|---|---|---|---|---|
| | Accuracy | Mean IoU | ΔAccuracy | ΔMean IoU | ΔAccuracy | ΔMean IoU |
| FCN-RN50 | 91.4 | 60.5 | 0.3 | -0.3 | 0.3 | 0.1 |
| FCN-RN101 | 91.9 | 63.7 | 0.2 | 0.0 | 0.3 | 0.4 |
| DeepLabV3-RN50 | 92.4 | 66.4 | 0.0 | 0.3 | 0.2 | 0.4 |
| DeepLabV3-RN101 | 92.4 | 67.4 | 0.1 | 0.5 | 0.5 | 0.9 |
| DeepLabV3-MobileNetV3-Large | 91.2 | 60.3 | -0.3 | -1.5 | 0.1 | 0.1 |
| LR-ASPP-MobileNetV3-Large | 91.2 | 57.9 | -0.4 | -0.9 | 0.1 | 0.2 |

### B.2  Object Detection

We use TorchVision (https://github.com/pytorch/vision/tree/main/references/detection) and its default hyperparameters to perform experiments on several networks performing semantic segmentation on the COCO2017 dataset, after pre-training the backbones with ILSVRC2012, shown in Table 6.

### B.3  Language Translation

We use the NVIDIA Deep Learning Examples (https://github.com/NVIDIA/DeepLearningExamples) and FairSeq (https://github.com/pytorch/fairseq/tree/main/examples/translation) and their default hyperparamters to perform experiments on GNMT and Transformer, respectively, using the WMT'16 EN-DE dataset, shown in Table 7.

Table 6: Object detection networks' bounding-box results improve under the same fine-tuning schedule when using permutations. (COCO2017)

| Network | Baseline (Dense) | | Default 2:4 | | **Permuted (ours)** | |
|---|---|---|---|---|---|---|
| | Precision | Recall | $\Delta$Precision | $\Delta$Recall | $\Delta$Precision | $\Delta$Recall |
| Mask R-CNN (RN50) | 37.9 | 51.8 | -0.1 | -0.1 | 0.3 | 0.7 |
| Keypoint R-CNN | 54.6 | 63.5 | 0.7 | 0.7 | 0.9 | 0.9 |
| RetinaNet RN50 FPN | 36.4 | 53.9 | -0.7 | -0.7 | 0.2 | 0.2 |

Table 7: Language translation networks' results improve under the same fine-tuning schedule when using permutations. (WMT'16 EN-DE)

| Network | Baseline (Dense) | Default 2:4 | **Permuted (ours)** |
|---|---|---|---|
| | BLEU4 | $\Delta$BLEU4 | $\Delta$BLEU4 |
| GNMT | 24.37 | 0.40 | 0.64 |
| FairSeq Transformer | 28.01 | 0.02 | 0.08 |

## B.4 Language Modeling

We use the NVIDIA Deep Learning Examples (https://github.com/NVIDIA/DeepLearningExamples) and its default hyperparameters to perform experiments on BERT-Large fine-tuned for the SQuAD task, both versions 1.1 and 2.0, shown in Table 8.

Table 8: BERT-Large's modeling capabilities (on both the SQuAD v1.1 and v.20 datasets) improve under the same fine-tuning schedule when using permutations.

| Dataset | Baseline (Dense) | Default 2:4 | **Permuted (ours)** |
|---|---|---|---|
| | F1 | $\Delta$F1 | $\Delta$F1 |
| SQuAD v1.1 | 91.35 | 0.13 | 0.30 |
| SQuAD v2.0 | 81.22 | 0.21 | 0.42 |

## C  Counting Unique Permutations

In Section 4.2, we presented $\frac{C!}{M!^G G!}$ as the number of unique permutations for a matrix with $C$ columns and a group width of $M$, leading to $C/M = G$ total groups. Here, we derive this limit, starting with the expanded form of the total number of ways to choose four columns from a matrix with $C$ columns, $\binom{C}{4}$, and solve when $C = 4$:

$$\frac{C!}{(C-4)!4!} = \frac{4!}{0!4!} = 1 \tag{1}$$

As expected, there's one way to choose four columns from a set of four columns. If we add another group (four channels) to the matrix, this first stripe now has

$$\frac{8!}{(8-4)!4!} = \frac{8!}{4!4!}$$

combinations. And, for each of them, the second (final) stripe has $\binom{4}{4}$ options. Multiplying and simplifying $\binom{4}{4}\binom{8}{4}$:

$$\frac{4! \times 8!}{0!4! \times 4!4! \times 2!} = \frac{8!}{4!^2 \times 2!} = 35 \tag{2}$$

Since the two stripes could have been in either order without changing of the N:M (2:4) pruning decisions, there is an additional divisor of 2!, the number of permutations of 2 stripes. Repeating this process once more for three total stripes:

$$\frac{4! \times 8!12!}{0!4! \times 4!4! \times 8!4! \times 3!} = \frac{12!}{4!^3 \times 3!} = 5775 \tag{3}$$

Generalizing to any number of columns, $C$, we have:

$$\frac{C!}{4!^{C/4} \times (C/4)!} \tag{4}$$

Of course, we are using 4 as the group width $M$ (number of columns in a group), and $C/4$ (or $C/M$ is simply the number of groups in C columns, $G$. Replacing the constants with these variables lets us generalize to:

$$\frac{C!}{M!^G G!} \tag{5}$$

This general formula builds in the two constraints for a unique permutation: (1) the order of the stripes does not matter, and (2) the order of columns within a stripe does not matter. Starting with the total number of permutations of $C$ columns, $C!$, (1) is satisfied by the $G!$ divisor. Each stripe is further limited in its uniqueness by the number of ways there are to order the values (column indices) in each stripe ($M!$) for each of $G$ groups, constraint (2), by the $M!^G$ divisor.

The algorithm described in Section 4.2, which generates all the unique permutations in O(n) time is shown in Listing 2.

---

**Algorithm 2:** Generate unique permutations [O(n)]

---

**Result:** global UniquePermutations

**def** *is_canonical_form(perm, col, M)***:**
    **if** *len(perm) % M == 0***:**
        **if** *cols < col not in perm***:**
            return False;
        return col > perm[-M];
    return col > perm[-1];

**def** *generate_permutations(perm, remaining, M)***:**
    **if** *remaining is empty***:**
        UniquePermutations = UniquePermutations.append(perm);

    **for** *col in remaining***:**
        **if** *is_canonical_form(perm, col, M)***:**
            generate_permutations( perm.append(col), remaining.remove(col), M);

**def** *generate_all_permutations(C, M)***:**
    init_perm = [0];
    remaining_channels = [c for c in range(1,C)];
    generate_permutations(init_perm, remaining_channels, M);

---

## D   Perfect Mask Discovery

The exhaustive search shown in Listing 2 *will* identify a perfect permutation ($p_p$) that has 100% efficacy, but we can reduce the cost of this exhaustive search by exploiting the property that a permutation with 100% efficacy removes the smallest 50% of values in each row. By first generating the optimal mask (by applying 50% unstructured sparsity to each row) and then permuting the *mask* until every group satisfies the 2:4 constraint, we do not need to either (1) compute sums of speculatively-pruned values or (2) traverse sub-optimal branches of the permutation tree. These optimizations are shown in Listing 3.

By first generating the perfect mask, incrementally building a permutation can trivially reject all recursively-generated permutations for which the permuted mask does not satisfy the N:M constraint. If a dense matrix has such a perfect permutation, this is a very efficient solution. If a dense matrix does not have a perfect permutation, this routine will quickly return no solution. If a *sparse* matrix is given as the input, though, it can be nearly as inefficient as a full exhaustive search, since there will be many plausible branches in permutation space that fail to satisfy the constraint close to the leaves of the tree, rather than closer to the root.

**Algorithm 3:** Find a perfect permutation (if it exists).

```
Result: global PerfectPermutation
def is_canonical_form(perm, col, M):
    if len(perm) % M == 0:
        if cols < col not in perm:
            return False;
        return col > perm[-M];
    return col > perm[-1];

def generate_permutations(perm, remaining, N, M, Mask):
    if len(perm) % M == 0 and not Mask[:,perm].satisfies(N,M):
        return;

    if remaining is empty:
        PerfectPermutation = perm;
        endProcedure();

    for col in remaining:
        if is_canonical_form_and_is_N:M(perm, col, N, M, Mask):
            generate_permutations( perm.append(col), remaining.remove(col), M);

def find_perfect_permutation(Weights, C, N, M):
    PerfectPermutation = None;
    PerfectMask = prune_rows(Weights, N/M) != 0.0;
    init_perm = [0];
    remaining_channels = [c for c in range(1,C)];
    generate_permutations(init_perm, remaining_channels, N, M, PerfectMask);
```

# E   Trading Minima with Stochasticity

There are other ways to induce stochasticity in a greedy search that will not necessarily improve the final efficacy. While these cannot help an already-converged search escape a minimum, they *can* help an in-progress search to bypass one minimum in favor of another. By repeating a search or construction with different random seeds and choosing the best result, these techniques may allow the search to find a new, better permutation:

- Random initializations, as one might use to try different weight initializations in training a neural network, can be analogous to first applying a random permutation before commencing the greedy search.

- Rather than traversing the swap map in greedy order, one could select entries at random from the set of swaps with positive improvements. By moving only in a positive direction, the search will still converge.

To show that it is necessary to repeat the search for all random seeds and that it is not sufficient to randomly search for an improved initialization and greedily search from *there*, Table 9 shows the experimental efficacy for three strategies on the same random input matrix: (1) the best of 10 random permutations, (2) a greedy search, and (3) the solution found by (1) followed by (2). While (1) improves over the input, it does *not* lead to a better final minimum.

Table 9: A better starting permutation does not guarantee a better permutation after searching.

| Strategy | Efficacy |
|---|---|
| (1) best of 10 random permutations | 7.5% |
| (2) greedy search | 46.5% |
| (3) (1) then (2) | 46.1% |

# F   Efficacy Details for MobileNet v2

When we fine-tune MobileNet v2 with the baseline permutation search algorithm, the top-1 and top-5 accuracies are still worse than the dense baseline by -1.26 and -0.69 percentage points, in contrast to using our improved algorithm with suggested settings, optimizing stripe groups of size 2 with 100 bounded regressions, which fully

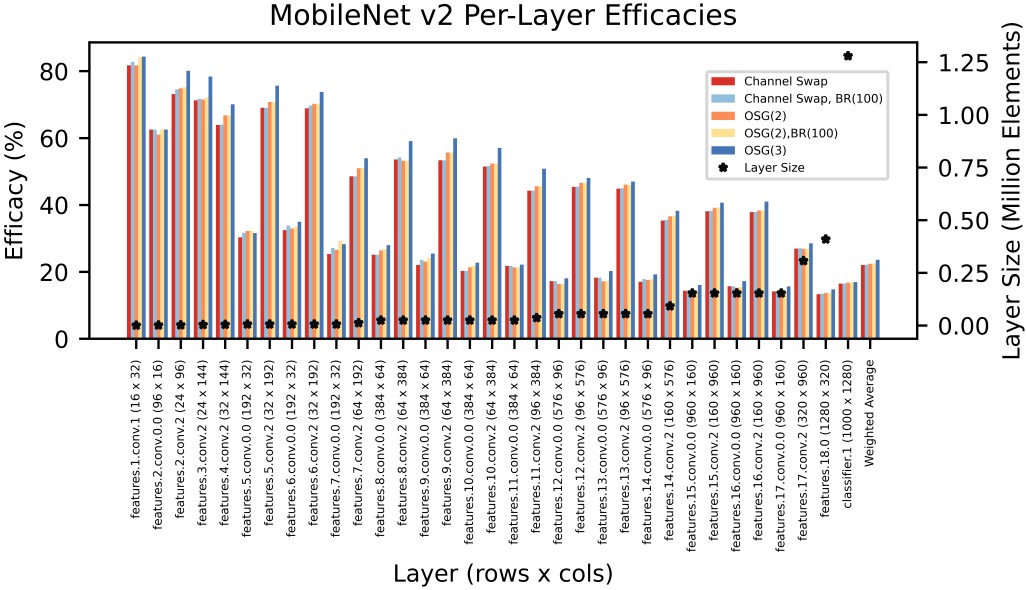

Figure 5: Per-layer efficacies for MobileNet v2 using different search strategies. Though the weighted average values are very similar due to the overwhelming influence of the classifier, the difference between the baseline algorithm ("Channel Swap") and our proposed setting ("OSG(2),BR(100)") is pronounced in individual layers.

recovers to the original accuracy with the same fine-tuning schedule. As shown in Figure 5, the weighted average efficacies over all layers for these two search techniques are within 0.4 percentage points. This single metric, weighted average efficacy, hides some details, though - in 17 out of 33 layers, the efficacy of the proposed algorithm is better by more than a full percentage point, and the non-weighted average improvement is similarly more than a full percentage point, despite some layers performing slightly better with the baseline approach. It is the small layers which show improvement, rather than the large classifier, which are likely responsible for the significant improvement in final network accuracy.

This detailed breakdown also shows interesting trends in efficacy:

- For the same number of rows, increasing the column count allows more effective permutations. This is because we have more degrees of freedom to change effective pruning groups.

- This effect is even more pronounced in layers with swapped row and column counts, where we simultaneously reduce the number of rows and increase the number of columns.

- For the same aspect ratio, increasing the element count reduces efficacy.

## G   The Importance of Small Improvements

It could seem that small improvements in efficacy may have only a minor effect on final network accuracy, especially considering the noisiness inherent in large-scale training. However, there is still practical value to be gained from relatively minor improvements in permutation efficacy. If a network has been pruned without structure, it may be advantageous to convert that network to a structured-sparse network for more efficient deployment. Permuting the weights can, as in the case of pruning a dense matrix, reduce the impact of imposing some N:M sparsity constraint, and make it more likely fine-tuning will recover accuracy (or reduce the amount of fine-tuning needed). Better than reducing the magnitude of lost weights, though, is completely eliminating it - by using the zeros already present in the unstructured sparse weight matrix, it may be possible to find a permutation that does not lose *any* magnitude after applying the N:M constraint. Small improvements in the permutation search strategy can make this possible.

As an example, we prune a layer from MobileNet_v2 (features.8.conv.2 with 384 channels and 64 1x1 filters) with unstructured sparsity at rates of 70% and 75% and show the efficacy of different strategies in Table 10 (with the same meaning as Table 1). 70% unstructured sparsity is insufficient to achieve 100% efficacy with the permutation strategies attempted; perhaps increasing the size of the stripe groups being optimized in the

Table 10: Even small improvements in permutation efficacy can have a large effect on the utility of a permutation - perfect permutations with efficacy of 100.0 mean that no fine-tuning is required for some layer. BR $(B)$ = bounded regressions (a total of $B$ pairs of channels are swapped when the greedy phase converges), OSG $(D)$ = optimize stripe groups of size $D$ during the greedy phase.

| Greedy Phase | Escape Phase | Efficacy 70% | 75% |
|---|---|---|---|
| Channel Swap | - | 95.2 | 99.7 |
| Channel Swap | BR (100) | 96.8 | 100.0 |
| OSG (2) | - | 95.8 | 99.8 |
| OSG (2) | BR (100) | 96.2 | 100.0 |
| OSG (3) | - | 99.5 | 100.0 |
| OSG (3) | BR (100) | 99.9 | 100.0 |

greedy phase could recover the final 0.1% efficacy. While 75% unstructured sparsity can give an efficacy of 100%, bounded regressions are required to escape the local minima and find the optimal solution for both the simple greedy phase as well as optimizing stripe groups of size 2 in the greedy phase; simply increasing the size of the stripe groups to 3 is enough to find the optimal permutation without requiring bounded regressions. In this case, the default approach (with no mechanisms to escape a local minimum) has an efficacy of 99.7%, but this is, behaviorally, a long way from 100%.