# OpenReview forum: "Channel Permutations for N:M Sparsity"
_NeurIPS.cc/2021/Conference — NeurIPS 2021 Poster_

### Official Review · Reviewer_rysJ · 2021-07-07

**Rating:** 7
**Confidence:** 5

**Summary:**

This paper introduces a channel permutation method to maintain the important connections. the idea is very interesting and meaningful.

**Limitations And Societal Impact:**

Weak points

the experiments only consider the classification.

**Main Review:**

Strong points:

This paper emphasizes the drawback of N:M sparsity when each group only has M elements.

To handle the local minima solutions, this paper proposes the bounded regression method, which is simple yet effective, to escape local permutation.

The channel permutation method does not incur inference cost and can handle the complex structure of DNNs, These points are important for sparse neural networks.

the experimental on ImageNet1K support the claims with various CNN-based models. This paper is a good practical work to facilitate the deployment of N:M sparse model.


Weak points

The experiments only consider the classification.


Overall, I am inclined to clear accept this paper.

**Time Spent Reviewing:**

7

---

> ### Author Response · Authors · 2021-08-06
> **Author Response (to rysJ)**
>
> Thank you for your feedback - we agree that one limitation of our work is that we have only presented results for image classification.  Our techniques for escaping the local minima found by the purely greedy process are task-agnostic, however; they will succeed for any weight matrix.  The question becomes: will permutations help other tasks?  We believe this should be the case, though, as long as either (1) the intuition that maintaining weight magnitude after pruning by magnitude leads to a more accurate network holds for those tasks, or (2) another pruning metric is identified that does have a more straightforward relationship between itself and network accuracy.
>
> We have early results for other tasks, including semantic segmentation, object detection, and language translation and modeling that show final 2:4 sparse accuracy is improved under the same fine-tuning, but these results were not available in time for submission.

---

> ### Comment · Reviewer_rysJ · 2021-08-18
> **Vote for acceptance**
>
> Thanks for the detailed response. I am very much looking forward to these changes in the paper.

---

### Official Review · Reviewer_fH7G · 2021-07-15

**Rating:** 6
**Confidence:** 5

**Summary:**

The paper focuses on fine-grained block sparsity setting (N:M sparse network) and in specific the case where a pre-trained model is given and one wishes to extract an N:M mask and retrain. The authors suggest using channel permutation to reduce the initial accuracy degradation and demonstrate a significant accuracy boost on parameter-efficient models using their method.  They suggest two greedy methods to find a good permutation. Additionally, the authors used their method to transform several layers from an unstructured sparse model to an N:M structure sparse layers.

**Limitations And Societal Impact:**

The authors discussed the potential social impact but did not address the limitation of their study.

**Main Review:**

Pros:

1) The main idea of the paper (channel permutation) is simple and seems to work.
2) While the idea is not novel as it was suggested before [1] for a different type of pruning it was never used for N:M sparsity.
3) The authors check their approach on several models.
4) The authors demonstrate that given enough resources their algorithm can find the optimal permutation.

Cons:

1) The paper is poorly written and hard to follow. There are many terms that were not defined or defined in the appendix without reference. For example, G in line 174 was defined only in the appendix and DGRAD in line 346 was not defined at all.
2) While channel swapping is clearly a combinatorial problem it was never discussed or formulated (is this an NP-hard problem?). I believe that finding the right formulation would enable using open-source parallel packages for combinatorial problems.
3) No code!  paper with no code has a much lower impact as other researchers would probably avoid using or extending it.

Questions:

1) While table 2 demonstrates that OSG is much better than channel swapping, and figure 3 demonstrates that weight average efficacy correlates with accuracy improvement. The authors did not present results with simple channel swapping. Thus, it is not clear why the OSG overhead is required.
2) OSG is clearly not a very light technique, can you comment on the amount of time it takes to find the best permutation (with OSG(3)-BR-1000, OSG(2) BR-1000, channel swap).
3) Have you tried to fix the bias as suggested by [2] perhaps it should help with “unstructured to structure”  experiments as suggested by [2]. I also believe you should mention [2] in this paragraph. Finally, it might help for “fine-tuning” experiments with parameter efficient models (table 2) as well.

[1] Ji et al., TETRIS: TilE-matching the TRemendous IrregularSparsity

[2] Hubara et al., Accelerated Sparse Neural Training: A Provable and Efficient Method to Find N:M Transposable Masks.

**Time Spent Reviewing:**

3-4 hours

---

> ### Author Response · Authors · 2021-08-06
> **Author Response (to fH7G)**
>
> Thank you for your helpful comments and feedback.  We agree that the idea of permutations is not novel, as they do appear in [1], but we show a deficiency in the solution proposed therein (local minima) and propose two techniques to improve its results.  So, we not only apply permutations to a new domain, but also improve upon the existing solution found for past domains.
>
> With your feedback, and that of other reviewers, we have several avenues to pursue to improve the clarity of our manuscript.  The definition of ‘G’ was unknowingly edited out for space, we’ll certainly add it back to the main content!  DGRAD is the standard “Data Gradient” computation phase of back-propagation.
>
> Finding a novel formulation for an existing solver seems like it could be fruitful future work.  Better permutations, or more efficient solutions, could make the approach tractable for very large matrices and more demanding use cases.
>
> We plan to release a package to allow practitioners to use our algorithms seamlessly in the near future, but it will not be available before the discussion period ends, unfortunately.
>
> ### Answers to questions
>
> 1. We did not perform the full fine-tuning for simple Channel Swaps (or any of the lesser techniques) in order to limit unnecessary training cycles.  Since we have shown that our techniques improve efficacy, and improved efficacy results in improved final network accuracy, accuracy with our approach *should* be strictly better.  With the margin of victory so close in some cases (0.01/0.02 percentage points for MobileNet v2, for example), we believe that our techniques are necessary in order to fully recover accuracy.  Further, the importance of OSG is quantitatively demonstrated in Figure 4: optimizing stripe groups (and using bounded regressions) is more successful at transparently transforming unstructured sparse weights to 2:4 sparse weights.  These two results suggest that OSG is required for the highest probability of success, whatever metric is used (final network accuracy or number of unstructured sparse layers converted to N:M sparse layers).  Finally, the overhead of OSG, compared to Channel Swaps, is not what you may think.  With our efficient implementations, OSG is able to find better solutions with fewer iterations than Channel Swaps, so the runtime is reduced compared to Channel Swaps.
>
> 2. As seen in our in-depth runtime table in our response to Reviewer GQXR, our default “OSG(2), BR(100)” is very fast for reasonable input sizes, and is faster than our implementation of “Channel Swaps, BR(100).”  We stress that while both implementations could likely be improved, we did not optimize one any more than the other.  Increasing the size of stripe groups optimized to 3 quickly increases the runtime, so we do not suggest it to be used for large weight tensors.  Using additional bounded regressions increases the runtime (sub-)linearly.
>
> 3. We have not tried to fix the bias, as we did not attempt to fine-tune the unstructured sparse networks.  Our goal was to map unstructured sparse weights into N:M structured sparse weights so they could be accelerated without performing any fine-tuning.  We agree: if fine-tuning is performed anyway, it is likely that the bias trick will work well with our permutations, both for unstructured to structured experiments as well as the results in Table 2, as you point out.  We mentioned this reference twice in our discussion about the unstructured to structured experiments ([11] in our submission); is there somewhere else you thought it should be included?
>
> ### Limitations
>
> We offered a discussion of what we see as the limitations of our work in the “Limitations” paragraph of the conclusion; are there other limitations we have not identified?
>
>
> [1] "Tetris: Tile-Matching the Tremendous Irregular Sparsity," Ji et al., NeurIPS 2018

---

> > ### Comment · Reviewer_fH7G · 2021-08-18
> > **Reply to authors response**
> >
> > Thank you for the response. You answered most of my questions. Below are my additional comments and clarifications:
> >
> > 1. Figure 3 demonstrates that weight average efficacy correlates with accuracy improvement up to 25% efficacy. All methods in table 1 have efficacy higher than 45%. Does the accuracy improvement platten for efficacy higher than 25%? If so perhaps the escape phase is not important. Can you please comment?
> > 2. Reading the paper and runtime table, it seems that the main advantage of OSG is its ability to optimize each group in isolation. This enables parallel optimization which allows you to inspect more permutations. Can you comment?  if this is true why didn't you mention it?
> > 3. I think that [1] was cited but [2] was not although they also suggest a light process that enables an unstructured sparse model to convert to an N:M structure model. While your method doesn't need any fine-tuning (which is great), its ability to speed up the inference is limited. This tradeoff (applying fine-tuning vs. inference speedup) can be discussed.
> > 4. I think that the main limitation is that you choose a suboptimal solution. Therefore it is not clear if: (1) an optimal solution would improve accuracy or (2) an even less optimal solution can achieve similar accuracies. I suggest adding it to the limitations paragraph.
> >
> >
> > [1] Ji et al., TETRIS: TilE-matching the TRemendous IrregularSparsity
> >
> > [2] Hubara et al., Accelerated Sparse Neural Training: A Provable and Efficient Method to Find N:M Transposable Masks.

---

> > > ### Author Response · Authors · 2021-08-18
> > > **Additional clarifications (fH7G)**
> > >
> > > Thank you for the opportunity to offer additional clarification:
> > >
> > > ### Efficacies
> > > Efficacies in Figure 3 cannot be compared directly to Table 1.  Table 1 is the efficacy on a single matrix (layer), while Figure 3 shows overall efficacy for entire networks, weighted by layer sizes.  So, while 64x128 matrices may see efficacies in the 40%-50% range, doubling both dimensions reduces efficacy by 10 percentage points (see our final table here: https://openreview.net/forum?id=WAO1STUPWPP&noteId=yX_wIWjsnJ).
> > >
> > > The aspect ratio of the matrix to be permuted also plays a role.  To give some intuition behind the range of efficacies seen in practice, EfficientNet_B0 has layers that range from tiny, with efficacies of 89.3% for channel swaps and 95.2% for OSG(2), BR(100), to short and wide (65.0%/65.9%), to tall and narrow (11.1%/12.3%).  Since larger layers are weighted more heavily in the weighted averages in Figure 3, and the larger the layer is for some aspect ratio, the lower the expected efficacy, the overall network efficacy is lower than the small matrices from Table 1.
> > > > Does the accuracy improvement plateau for efficacy higher than 25%?
> > >
> > > We do not know!  The range of efficacies we've reported in Figure 3 include the highest efficacies we achieved for those networks, using OSG(2), BR(100).  Generating data points higher on the efficacy axis will require further improvement on our techniques, such as some proposed in our future work (making optimizing larger stripe groups more efficient).
> > >
> > >
> > > ### Performance of OSG
> > > The advantage of OSG isn’t its ability to parallelize - an efficient implementation of channel swaps (including ours) will also be parallel.  After all, each group of M columns is independent from every other group, so when swapping two channels from two different groups, the implementation can check every pair of columns in those two groups while some other work unit works on other pairs of columns in other groups in parallel.
> > >
> > > Rather, the advantage of OSG is that, while the parallel work unit is performing work, it doesn’t just swap two columns at a time; it will look at all permutations available and potentially make a change that swaps more than two columns, making more “progress” towards convergence at each greedy step.  However, there may be optimizations we have not considered for the baseline channel swapping approach; our purpose in sharing runtimes was merely to show that OSG is not as time-consuming as might be assumed.
> > >
> > > (As described in the submission, the non-runtime advantage of OSG is that it adds depth to the permutation search, and can find solutions further away than simply swapping two channels.)
> > >
> > >
> > > ### Reference [2] / [11]
> > > > I think that [1] was cited but [2] was not although they also suggest a light process that enables an unstructured sparse model to convert to an N:M structure model.
> > >
> > > There may be a misunderstanding about the inclusion of Hubara et al’s work: your [2] is our [11], and we cited it twice in our unstructured->structured experiment paragraph (lines 307 and 311).  We agree, though, that fine-tuning after permutations (as was done in our other experiments, and mentioned on line 311 in the context of [11]'s work in fine-tuning unstructured sparse workloads), including using the bias trick from [2/11], can allow for accelerating more unstructured sparse layers if training data is available.
> > >
> > >
> > > ### Optimality
> > > > I think that the main limitation is that you choose a suboptimal solution.
> > >
> > > So we understand your point about optimality, are you referring to the optimality of the proposed algorithm itself, or the solutions found by the proposed algorithm?
> > >
> > > If the former, it is not clear to us that we chose a suboptimal solution - does an optimal solution exist? Does a *better* solution exist?  We’d love to hear about them, if so!
> > >
> > > If the latter, then this is fundamental to the nature of greedy algorithms.  While we have not proven it, we believe that determining optimality of a solution to the N:M permutation problem is only possible if the optimal solution can be found in every case.  That is, unless we know the optimal solution, we cannot tell if an arbitrary solution is sub-optimal.  We can add this to our limitations, but we do point out in our future work that there exist opportunities for finding better solutions with more efficient computation.
> > >
> > > > Therefore it is not clear if ... an even less optimal solution can achieve similar accuracies.
> > >
> > > We believe we have answered this question as well as it can be answered without defining "similar." Figure 3. indicates that an inferior solution would have lower final accuracy, as accuracy keeps improving up to our final, highest permutation efficacy, for both networks shown. Whether or not these lower accuracies are "similar" is difficult to answer: how much worse can something be before it is no longer similar?
> > >
> > > One question would be, though, why does it matter?  Given the choice between two solutions with similar costs (in our case, we're being generous with the term, since our solution is faster than our implementation of the worse solution, including all the optimizations described in the original publication; so, let us assume that the two solutions take the same time), is there any reason to use the inferior solution over our proposed improvements?
> > >
> > >
> > > ### Not just efficacy/accuracy
> > > We'd like to stress that advantages over past work are not just improved efficacy/accuracy.
> > > 1. The baseline solution requires the insertion of new operations into the network graph to maintain consistency between layers.  Ours is the first such technique that handles all reordering as an offline process by virtue of permuting only along one dimension.  Even if past techniques matched efficacy, they also require runtime overhead.
> > > 2. Our technique exposes tradeoffs: for small to medium layers, optimizing stripe groups of size 3 or 4 is possible.  Similarly, if some runtime budget for the permutation search phase has not been exhausted, allowing more bounded regressions may find a better solution.  Our "default" proposal of OSG(2), BR(100) is both fast enough and useful for all layer sizes we've encountered, so we focus on it for this initial work.  In contrast, past work has no such flexibility.
> > >
> > > [2/11] Hubara et al., Accelerated Sparse Neural Training: A Provable and Efficient Method to Find N:M Transposable Masks.

---

> > > > ### Comment · Reviewer_fH7G · 2021-08-19
> > > > **Reply to Additional clarifications**
> > > >
> > > > Thank you for your detailed response.
> > > >
> > > > Q1 and Q4: I was not aware that efficacy can vary from 11% to 90% and that on average it is 25%.  I believe you should discuss this in the main paper and perhaps add a table of the weighted efficacy for an entire model with different methods.
> > > >
> > > > Q2. Thanks for clarifying, yet I can't seem to understand how OSG is so much faster than channel swapping. Eventually, for each permutation, you need to check the magnitude change. I assume OSG converges faster which would be better understood if you would add the final efficacy to the runtime table.
> > > >
> > > > Q3. My apologies, thank you for clarifying.
> > > >
> > > > Optimality: In your ablation study you demonstrated that OSG(2) BR(100) can find the optimal permutation to only 60% of the matrices. The readers might wonder if this is important (i.e. affect accuracy). If  OSG(3) BR(1000) which always finds the optimal permutation is not necessary (per your results in table 2) maybe we can just use OSG(2) without the escape phase. If you will state OSG(2) with and without the escape phase (e.g., BR-100) weighted average efficacy on SqueezeNet v1.0 (as suggested above) the readers can better understand the importance of the escape phase.

---

> > > > > ### Author Response · Authors · 2021-08-20
> > > > > **Further thoughts on optimality and its necessity**
> > > > >
> > > > > ### Efficacy Variation
> > > > > The difference in efficacies seen by different layer shapes and sizes is fascinating!  We may not be able to make space to fit a full network’s table into the main paper, but we can certainly add a brief discussion, with more details in the supplementary material.
> > > > >
> > > > > ### Runtime
> > > > > Unfortunately, we can’t answer this question at the moment.  An easy answer could be that we have missed an opportunity for optimization of the channel swapping approach, though our implementation and adaptation for 2:4 sparsity is roughly in line with stated performance of the baseline algorithm in past work.
> > > > >
> > > > > However, we believe that this consideration is secondary to our main contributions, which include improvements upon this baseline algorithm.  Since reviewers had well-justified concerns that these improvements may make the practical use of the technique prohibitively expensive, we shared runtime information for a few layer sizes to show that this is not the case - the permutation search runtime is a drop in the bucket of overall network training time, even with the extra complexity of our improvements.
> > > > >
> > > > > ### Optimality and Utility of Bounded Regressions
> > > > > Thank you for the extra clarifications - we understand your concern about optimality, now.  Let us first clarify the optimality of OSG(3), BR(1000): this setting has been shown to find the optimum permutation for only the inputs in Table 1.  For different or larger matrices, this may not be the case.  So, even if we used these settings for the network-level experiments, it's likely the optimum permutations would not be found for the larger matrices.  OSG(X), where X is the number of groups in the matrix, *would* yield the optimal solution, but this is often not tractable.  This would be preferred, but we show that bounded regressions are a less-costly, but still fruitful, method of improving upon the solution found by OSG(Y) (where Y << X).
> > > > >
> > > > > We agree that bounded regressions may not be necessary to fully recover some network’s accuracy given some fine-tuning schedule.  In fact, we provide evidence of this in the final three entries of Table 2 - for these networks, no permutations were required to maintain accuracy.  However, just as it would be wrong to conclude from these results that permutations are never required, it would be wrong to see a result showing that bounded regressions are not required and conclude that they are never helpful or required.  This information would not answer the question of “maybe we can just use OSG(2) without the escape phase:”
> > > > > 1. Other networks that we have not tested may require bounded regressions given the same fine-tuning schedule.
> > > > > 2. Other objectives, such as transforming unstructured to structured sparsity, reducing the amount of fine-tuning required, or training with sparsity from scratch, may require bounded regressions to be successful.
> > > > >
> > > > > Attempting to draw conclusions from the limited selection of results about which techniques are required for some task may mislead the reader into generalizing improperly rather than provide useful answers or guidance.
> > > > >
> > > > > With that caveat in mind, the difference in efficacy between OSG(2) and OSG(2), BR(100) for full dense networks is small: between 0.1 and 0.2 percentage points (e.g., 25.8% and 25.9%) for SqueezeNet v1.0, EfficientNet B0, and MobileNet v2.  Whether or not the difference is small enough to hide within the 0.01/0.02 percentage points of top-1/top-5 accuracy for MobileNet v2, for example, is impossible to say without fine-tuning.  It is likely that SqueezeNet v1.0 would recover its accuracy without bounded regressions.  (We're being strict
> > > > >
> > > > > While we lack time to fine-tune these networks, we *can* provide results for transforming unstructured sparsity to structured sparsity in a data-free scenario (no fine-tuning) that shows the importance of bounded regressions.  This is a subset of the data from Figure 4 in the submission, with new rows added for OSG(2), and OSG(2),BR(1000), in which each data point is the number of layers of ResNet-50 that can satisfy 2:4 sparsity if the entire network were pruned to some unstructured sparse target (here, from 78% to 96%):
> > > > >
> > > > > | Strategy \ Sparsity (%) | 78 | 80 | 82 | 84 | 86 | 88 | 90 | 92 | 94 | 96 |
> > > > > |:-------------------:|:--:|:--:|:--:|:--:|:--:|:--:|:--:|:--:|:--:|:--:|
> > > > > |    Channel Swaps    |  3 |  6 |  6 | 11 | 12 | 17 | 23 | 36 | 49 | 53 |
> > > > > |        OSG(2)       |  3 |  6 |  6 | 12 | 13 | 18 | 24 | 38 | 49 | 53 |
> > > > > |   OSG(2), BR(100)   |  5 |  6 |  7 | 12 | 14 | 19 | 26 | 40 | 50 | 53 |
> > > > > |   OSG(2), BR(1000)  |  6 |  6 | 10 | 12 | 16 | 22 | 26 | 45 | 51 | 53 |
> > > > > |        OSG(3)       |  6 |  6 | 12 | 12 | 15 | 22 | 33 | 47 | 51 | 53 |
> > > > >
> > > > > There are several sparsities at which using bounded regressions, or increasing the number of bounded regressions, allows more layers to be transformed successfully.  While they may or may not have been required for some of the networks in Table 2, they are unequivocally helpful in this case.
> > > > >
> > > > > Thank you, sincerely, for this opportunity to discuss our work and find which aspects deserve further treatment!

---

> > > > > > ### Comment · Reviewer_fH7G · 2021-08-29
> > > > > > **Vote of acceptance**
> > > > > >
> > > > > > I thank the reviewer for their detailed and honest response, I have no further questions.
> > > > > >
> > > > > > As stated, it is hard to understand from the limited selection of results if, and in which case, the suggested techniques are required. Yet. since channel permutation seems light and effective I believe that practitioners would try applying it to boost accuracy (given a publicly available code).
> > > > > >
> > > > > > I think this work has some novel and interesting ideas . Still, to better clarify the limitations of this work, I kindly ask the authors to add to their discussion\appendix the additional results detailed above

---

### Official Review · Reviewer_GQXR · 2021-07-16

**Rating:** 7
**Confidence:** 5

**Summary:**

Recent nVidia chips support hardware acceleration of structured sparsity patterns in weight matrices — specifically 2:4 sparsity where consecutive groupings of 4 entries have half of their entries set to zero.  The naive way to exploit this acceleration in deep learning is to prune weight matrices so as to meet this sparsity requirement, but doing this naively can be suboptimal if some groups contain many “important” weights and other groups contain all zeros already — it would be better if the “important” weights had a better distribution across groups.  So this paper proposes a way to permute the weights such that the pruning yields less loss.

The main idea is that if we can permute the weight rows/cols appropriately, then we can force pruning to give something that has more weight magnitude. The authors introduce permutation efficacy (and show on Imagenet experiments that this makes sense and acts as a kind of proxy for quality) and present a (mostly) greedy algorithm for optimizing this measure.  Final experiments on a number of Imagenet trained architectures show accuracy improvements over the naive method.


**Limitations And Societal Impact:**

Yes.

**Main Review:**

There are several strong points in favor of this proposed method.  First, it does not change running time for inference — (so the benefits are free in some sense) and does not require retraining or fine tuning.  The proposed method is also fairly intuitive (seems natural).  And finally it yields positive results in terms of accuracy — and according to the experiments seems generally applicable across a range of networks.

The weaknesses of this paper is that there is limited upside (the improvements seem to be quite small on average). There is also no discussion of running time of the optimization (I agree this time can be amortized away, but I believe it should be discussed). Finally, exposition needs improvement — I found a number of statements to lack in mathematical rigor (more details below) and some of the figures to be confusing which increased the cognitive burden of reading this paper.

Overall, though I find the paper to be somewhat incremental - I believe it’s worth sharing this with the community so I lean towards acceptance.


More detailed comments:
* It’s presented starting from the beginning as if it is obvious that preserving weight magnitude corresponds to good final accuracy, but this is not actually obvious (and I believe the authors recognize this in the experiments). It would be good to have a more transparent discussion about this issue at the top of the paper.
* The proposed method ignores the fact that the output channels can also be swapped and intuitively, optimizing jointing for swapping both the rows and columns of a matrix, one could achieve even greater efficacy — I wonder if it would be worth commenting on this possibility somewhere (or tell me that I’m wrong).
* Several times, the authors mention “applying the 2:4 constraint” without explicitly spelling out what this means exactly — does this just mean, dropping the lowest magnitude 2 weights in each group of 4 consecutive entries?  Or is there something fancier happening here?
* The Figure 2 example is very confusing to read — I would recommend that the caption identify the colors, label what every arrow represents and explain the permutation notation (e.g. does [1,0,2] mean that we permute row 0 to be first, row 1 to be 0th, etc?).
* More on this point, permutations are never really defined and this causes problems throughout.  At some point for example, the authors say that “a permutation is unique if its stripes channels are in sorted order…” — but what are the stripes of a permutation (I think I can infer, but it would be best to not require the reader to have to)?  I recommend stating upfront, that each weight matrix is associated with a permutation, saying how the permutation is notated and what index spaces it maps between.
* Table 1 ablation: given that efficacy is not clearly connected to final preserved accuracy, why not just report preserved accuracy here? Also I recommend reporting running times — it seems like the more exhaustive the search, the better optima the algorithm was able to find (which is not surprising). What if we were to compare against a random baseline where we just evaluate thousands of random unique permutations and take the best?
* Figure 3: report original dense performance/accuracy?
* Line 141: “to keep the network graph consistent” — don’t you mean to keep the parameterized function equivalent?
* Final minor thing: this pdf file is 38 Mb which is huge given that there are no graphics in this pdf and I don’t know if a printed screenshot was submitted instead of an original pdf… but it’s also not possible to search the document which made reading hard if I had to do a search within the document for a particular term.




**Time Spent Reviewing:**

2

---

> ### Author Response · Authors · 2021-08-06
> **Author Response (to GQXR)**
>
> Thank you for your in-depth review and comments.  Before addressing your detailed comments, we must first clarify one misunderstanding: our presented results *do* use fine-tuning; we use the same fine-tuning procedure as in [1], which itself exposed lost accuracy in small networks.  While the accuracy before fine-tuning could be improved after using channel permutations, we measure our success in the fine-tuned accuracy, which fully recovers for these networks, in contrast to the fine-tuned networks without channel permutations.  (The exception to fine-tuned results is our transforming unstructured sparse weights to N:M structured sparse without fine-tuning, the final result in our experiments.)
>
> ### Quantitative Benefit
>
> With respect to the perceived upside, we look at it this way: while the accuracy improvements in some cases seem small (between 0.44 and 4.61 - the high end of the range is *not* small!), the real impact is the full recovery of the baseline accuracy.  If a network loses accuracy with some technique, be it fine-grained sparsity, channel pruning, quantization, or otherwise, the practitioner must decide if the benefits of the simplification technique outweigh the lost accuracy.  Perhaps there is a different model altogether that would see a similar runtime but achieve the required (dense) accuracy?  On the other hand, if there is no accuracy loss after simplification, this decision is not required: if the dense model’s accuracy was acceptable, then a simplified model that promises better throughput, latency, or energy use with the same accuracy is strictly better.  Unfortunately, without channel permutations, this cannot always be achieved for small networks, as we documented in Table 2.
>
> Additionally, we see utility for channel permutations in other areas: we demonstrated that our techniques enable unstructured sparse layers to be accelerated with 2:4 structured sparse accelerators at lower sparsities than without our techniques, and we suggest that permutations may allow for more efficient training of sparse-from-scratch networks, etc.  In these cases, the upside is harder to quantify and needs further study.
>
> ### Runtime
>
> We implemented efficient, but not highly optimized, versions of both channel swaps and our proposed algorithm in CUDA. The largest contributor to the runtime of the permutation search process is the number of channels in a layer and the size of stripe groups used by the OSG greedy phase; other factors have linear impact on runtime.  Using a V100 accelerator (as we used in our training results), we vary the number of columns and rows (channels and filters) of a random matrix and report the time, in seconds, required for a number of search strategies (note that our default option, which was used for all network accuracy results, was **OSG (2), BR (100)**):
>
> |           Runtime   (s)      |  |    |  |   |  |   |   |  |
> |:---------------------:|:----:|:----:|:----:|:-----:|:-----:|:------:|:------:|:-----:|
> | ***Matrix Size***              |        |        |       |         |          |            |           |           |
> |        Columns        |  32  |  32  |  64  |   64  |  128  |   128  |   256  |  2048 |
> |          Rows         |  32  |  64  |  64  |  128  |  128  |   256  |   256  |  2048 |
> | ***Strategy***             |    ***Results:*** |     |        |          |       |            |         |           |
> | Random (1000)         |  1.1 |  1.0 |  1.2 |   1.4 |   2.0 |    3.3 |    6.1 |  90.7 |
> | Channel Swap, BR(100) |  0.1 |  0.1 |  0.3 |   0.2 |   1.0 |    0.9 |    5.1 | 601.2 |
> | OSG (2)               | ~0.0 | ~0.0 | ~0.0 |  ~0.0 |  ~0.0 |   ~0.0 |    0.1 |  40.1 |
> | **OSG (2), BR (100)**     |  0.1 |  0.1 |  0.1 |   0.2 |   0.3 |    0.4 |    0.8 |  79.6 |
> | OSG (2), BR (1000)    |  1.0 |  1.0 |  1.4 |   1.6 |   2.8 |    3.3 |    7.1 | 435.9 |
> | OSG (3)               |  0.1 |  0.2 |  0.4 |   0.9 |   8.9 |   22.8 |  130.7 |     - |
> | OSG (3), BR (100)     |  3.6 |  6.3 |  8.5 |  15.0 |  70.0 |  137.4 |  638.8 |     - |
> | OSG (3), BR (1000)    | 35.7 | 61.7 | 79.2 | 143.4 | 594.3 | 1129.0 | 4936.7 |     - |
>
> We can include a broad characterization of runtimes in the manuscript, but these details are likely too verbose to fit, and also not indicative of potential runtimes from a more optimized implementation.
>
> ### Detailed Comments
>
> - **Magnitude:** you are correct: we assume that increased magnitude will lead to higher accuracy; this stems from the pruning approach used by [1], which uses weight magnitude to choose which weights to prune.  We verify this assumption empirically in Section 5 (higher efficacy, or maintained weight magnitude, leads to higher network accuracy), but we also note that our technique can be used with any metric of interest, not just magnitude.  This is certainly worth an explicit discussion and will add it to the introduction.
> - **Permute both rows and columns**: this is not necessary for N:M sparsity.  Permuting the rows of some matrix does not change the pruned version of that matrix (other than having had its rows permuted) - that is, the pruning decisions in each group of M values are the same.  The composition of the N:M pruning groups are only affected by a permutation of columns.  It is this simplicity that allows us to apply permutations in a completely offline manner, with no runtime overhead.  Past work has relied on permutations in more than one dimension, which requires extra operations in the network graph to maintain consistency.  As we note in line 114, past work could use this approach, as well, to eliminate the runtime overhead, but as their applications do make use of permutations in multiple dimensions, it may lead to inferior permutations; of course, the techniques we introduce to escape local minima may help offset that reduced efficacy.
> - **Clarify**
>     - Applying the 2:4 constraint is exactly as you surmised: dropping the smallest two values out of every four.  We will make this clear.
>     - Your suggestions for the caption of Figure 2 are excellent, thank you.
>     - We’ll also clarify what we mean by a permutation, as well as what a “stripe” is in the context of an N:M sparse matrix
> - **Ablation study**: performing the ablation study on full networks would require orders of magnitude more computing resources than we currently used, and the single matrices we used for the experiments in Table 1 do not have a notion of accuracy, since they are matrices of random weights, not part of a trained network solving a task.  The runtimes above will give you an idea of the runtimes for these matrix sizes.  We also test the “random permutations” experiment for 1000 permutations and compare its efficacy with that of our default algorithm, OSG(2), BR(100), below.  It not only takes longer to run, but also results in lower efficacies: in our experience, it is easy to find a poor solution, but very hard to find a good one!
> - **Original accuracy in Figure 3**: The point of figure three is not to show that we have matched the baseline accuracy, or any other target, just that there is empirical correlation between our efficacy metric and final network accuracy.  We think this conclusion is easier to draw without the extra information, though it may be interesting to more curious readers.  Please let us know if you think it would add significant value to the figure, though.
> - **“Network graph” vs. “parameterized function:”** this substitution seems appropriate and correct, thank you for pointing this out.
> - **Large, unsearchable PDF**: this is a terrible oversight!  It seems to be a product of the method we used to split the submission into a main paper and supplementary material; the un-split file is only 1MB and is searchable.
>
> |           Efficacy      |  |    |  |   |  |   |   |  |
> |:---------------------:|:----:|:----:|:----:|:-----:|:-----:|:------:|:------:|:-----:|
> | ***Matrix Size***              |        |        |       |         |          |            |           |           |
> |        Columns        |  32  |  32  |  64  |   64  |  128  |   128  |   256  |  2048 |
> |          Rows         |  32  |  64  |  64  |  128  |  128  |   256  |   256  |  2048 |
> | ***Strategy***             |    ***Results:*** |     |        |          |       |            |         |           |
> | Random (1000)         |  41.7 |  29.3 | 17.8 |  11.2 |  8.8 |  6.1 |  2.8 |  0.2 |
> | **OSG (2), BR (100)**     |  63.5 |  50.0 | 49.0 |  35.5 |  35.8 |  25.5 |  26.7 | 11.4 |
>
> [1] "Accelerating Sparse Deep Neural Networks," Mishra et al., 2021

---

> > ### Comment · Reviewer_GQXR · 2021-08-31
> > **Response to authors**
> >
> > Thanks - this response helped to clear up some concerns.  I've slightly bumped up my original rating to reflect this --- and am looking forward to the final version.

---

### Official Review · Reviewer_32Qz · 2021-07-25

**Rating:** 8
**Confidence:** 4

**Summary:**

In this paper, the authors proposes channel permutation to improve the accuracy of N:M sparsified networks. Compared with previous works on N:M sparsity, channel permutation could maximize the magnitude of the weight matrix after N:M sparsification. Thus, the accuracy of the sparsified network could be preserved. To find the optimal permutation, a greedy searching algorithm is proposed, which is enhanced by two local minima escaping techniques including "bounded regressions" and "narrow, deep search". The effectiveness of the proposed channel permutation and local minima escaping methods are validated by experiments on a sufficient number of network architectures.

**Ethics Review Area:**

["I don’t know"]

**Limitations And Societal Impact:**

The authors have addresses this problem properly.

**Main Review:**

Pros:
1. The idea of channel permutation for N:M sparsity is quite novel.
2. The problem of finding the optimal channel permutation is well addressed by the greedy search and the local minima escaping technique.
3. The experimental results validates the benefits brought by the proposed method.

Cons:
1. Some of the details in the paper is not well described.
    Line 160: It is very clearly how the term $W_{50\%\_rows}$ is defined. Is the 50% sparisty constraint applied to each row of the matrix independently?
2. Missing references to recent work on network pruning.
[1] https://arxiv.org/abs/1903.10258
[2] https://arxiv.org/abs/2003.08935
[3] https://arxiv.org/abs/2006.12813



**Time Spent Reviewing:**

3

---

> ### Author Response · Authors · 2021-08-06
> **Author Response (to 32Qz)**
>
> Thank you for the helpful comments and feedback.  We agree that the exposition could be more clear, and we have been working to improve this.
>
> To answer your particular question: yes, the 50% sparsity constraint is applied to each row independently.  If we applied 50% sparsity to the entire matrix, there could be some row that does not have any values removed; if that were to be the case, then there is no channel permutation that could keep that row from losing extra values after applying the 2:4 constraint.  So, when determining a upper bound for permutation efficacy, we must apply the 50% unstructured sparsity to each row.  This is not sufficient to guarantee that there is a single permutation that perfectly preserves all the values in each row, though, so this is a loose upper bound.
>
> The references you suggest will complement those we included for channel/filter pruning nicely, thank you.

---

> > ### Comment · Reviewer_32Qz · 2021-08-21
> > **Reply to author response**
> >
> > Thanks for the response to my question about the definition of the 50% sparsity and missing references. Looking forward to the revised version of the paper.

---

### Decision · Program_Chairs · 2021-09-27

**Decision:**

Accept (Poster)

**Comment:**

This paper suggests an interesting permutation based method which closes the (sometimes small, but significant) performance gap in 2:4 fine-grained sparsity. The reviewers were all positive, and it seems most of their clarity-related concerns were addressed in the authors' response and the following discussion. I ask the authors to address all these issues in the camera-ready version to improve the readability of the paper and publish the code, as promised.